# Star-DS: Step-level Uncertainty-Aware Reasoning Data Selection in Reinforcement Learning for LLM Multi-step Reasoning

## Abstract

Large language models have demonstrated remarkable potential on complex multi-step reasoning tasks, largely enabled by substantial post-training via reinforcement learning with process reward verification on reasoning datasets. Recent studies have shown that it is possible to alleviate the massive data reliance and computational costs by selecting high-value subsets of data while maintaining reasoning capability. However, existing data selection methods typically rely only on outcome-level signals derived from final answers to measure data quality, overlooking step-level signals that are intrinsic to multi-step reasoning, which leads to suboptimal identification of valuable reasoning data. In this paper, we propose a novel **S**tep-level Uncertain**t**y-**A**ware **R**easoning **D**ata **S**election approach (**Star-DS**) that incorporates both step-level and outcome-level signals for identifying high-value reasoning data in reinforcement learning for LLM multi-step reasoning. Specifically, we introduce step-wise self-evaluation uncertainty of each reasoning step, as well as reward variance of the final answer, to quantify the value of each sample for RL training. Experiments with diverse reasoning models across multiple benchmarks demonstrate that our approach consistently identifies high-value data, preserves multi-step reasoning performance after RL training, and significantly reduces both data requirements and computational costs. [1]

## 1 Introduction

Recent advances in large language models (LLMs) have shown that reinforcement learning (RL) is a highly effective post-training paradigm for enhancing complex reasoning abilities Rafailov et al. (2023; 2024); Shao et al. (2024). A key factor underlying these gains is the ability of LLMs to generate step-by-step solutions in a chain-of-thought (CoT) format (Wei et al., 2022), which allows them to perform multi-step inference and tackle a wide range of complex tasks. Models such as o1 (Jaech et al., 2024), DeepSeek-R1 (Guo et al., 2025), and Kimi1.5 (Team et al., 2025) empirically demonstrate that RL fine-tuning on CoT outputs can elicit sophisticated reasoning behaviors, including reflection, self-verification, and extended reasoning chains.

Prior studies (Gao et al., 2025; Li et al., 2025b; Wang et al., 2025) have emphasized that, during the post-training stage of reasoning LLMs, the quality of training data often plays a more crucial role than its sheer quantity. In certain scenarios, a carefully curated small subset of high-quality data can achieve comparable performance to training on the entire dataset (Wang et al., 2025), consistent with the principles of few-shot learning. Motivated by this observation, several methods have been proposed to improve data selection for model optimization and efficiency. For instance, Learning Impact Measurement (LIM) (Li et al., 2025b) identifies samples whose learning patterns complement the model's overall performance trajectory, demonstrating their potential value for training. Gradient alignment-based approaches (Li et al., 2025a) estimate the influence of individual data points on the training loss, offering a theoretically grounded framework for data selection. GRESO (Zheng et al., 2025) predicts and skips zero-variance prompts using reward training dynamics before the rollout stage, further reducing computational overhead during RL training. Other strategies leverage either

---

[1]Code is available at `https://anonymous.4open.science/r/Star-DS-5961`

an external expert model or the learning signals of the target model to rank and select high-value samples.

While these methods offer valuable insights and are effective in certain RL scenarios, they may not fully capture the fine-grained dynamics of reasoning, which can lead to suboptimal performance. This is largely because these methods solely focus on rollout outcomes. For example, LIM and GRESO rely on aggregate measures such as final answer correctness, while gradient-alignment techniques estimate the influence of data points through the loss function. None of these methods is specifically reasoning-oriented, as they often overlook the rich information contained in intermediate reasoning steps, which are intrinsic to multi-step inference and crucial for reasoning tasks. These gaps highlight the need for a data selection methodology explicitly designed for reasoning-oriented LLMs, capable of improving both efficiency and performance in multi-step inference contexts.

Addressing the aforementioned gaps poses a key challenge: how to leverage step-level learning signals within rollouts for data selection while effectively combining them with outcome-level signals. To tackle this, we draw inspiration from uncertainty estimation techniques of LLMs (Zhang et al., 2025; Vashurin et al., 2024). The central intuition is that the uncertainty or inconsistency of a model on a given input serves as a natural indicator of the sample's potential value for training (Zhao et al., 2025; Fu et al., 2025). Samples where the model exhibits high uncertainty are those in which reasoning is unstable, diverse, or conflicting across different rollouts, and thus provide richer learning signals. Existing methods that consider outcome-level information, such as the divergence among final answers or deviations of performance trajectories from their mean, can be interpreted as a form of uncertainty. By extending this concept to step-level uncertainty (Ye et al., 2025; Kadavath et al., 2022), we can evaluate the reliability of intermediate reasoning steps, capturing fine-grained dynamics that outcome-level signals alone cannot reveal. Crucially, combining step-level and outcome-level uncertainty yields a more comprehensive measure of a sample's value, enabling more effective data selection that is both reasoning-oriented and uncertainty-informed.

In this paper, we propose the first data selection approach specifically tailored for multi-step reasoning in LLMs, namely **Step-level Uncertainty-aware Reasoning Data Selection (Star-DS)**. Our method is designed to leverage both step-level and outcome-level signals to identify high-value samples for reinforcement learning in multi-step reasoning tasks. Concretely, the step-level uncertainty is estimated via a self-evaluation mechanism, in which the model itself assesses the likelihood of correctness for each intermediate reasoning step. The outcome-level uncertainty is quantified using the standard deviation of rewards across different rollouts, capturing the variability of final answers. By aggregating these two complementary signals, our method provides a unified, uncertainty-aware metric for reasoning data selection, allowing the model to prioritize samples that are both informative and learnable for training. Our experiments demonstrate the effectiveness of the proposed method in improving both reasoning performance and training efficiency.

**Contributions.** Our main contributions are as follows:

- We introduce the first data selection method in RL designed explicitly for LLM multi-step reasoning, overcoming the shortcomings of previous approaches that rely solely on rollout outcomes and neglect the informative process signals from intermediate reasoning steps.

- We propose a Step-level Uncertainty-aware Reasoning Data Selection method that integrates step-level self-evaluation uncertainty with outcome-level reward variability to effectively identify high-value samples for reinforcement learning on multi-step reasoning tasks.

- Extensive experiments across diverse reasoning benchmarks, datasets, and models demonstrate that our approach improves both reasoning performance and training efficiency, and additional analyses reveal the complementary roles of step-level and outcome-level signals in data selection.

## 2 RELATED WORK

**Data Selection for LLM Reinforcement Learning.** Recent works have highlighted that the efficiency of reinforcement learning for reasoning LLMs critically depends on the quality of training data (Gao et al., 2025; Wang et al., 2025; Li et al., 2025b; Muennighoff et al., 2025). To this end, a wide range of data selection methods (Muldrew et al., 2024; Liu et al., 2024b; Das et al., 2024; Fatemi et al.,

2025) have been proposed to identify high-value training subsets, aiming to enhance the reasoning ability of LLMs while reducing the cost of large-scale data collection and training.

Learning Impact Measurement (LIM) (Li et al., 2025b) evaluates the utility of samples by estimating their contribution to the model's learning trajectory, aiming to prioritize examples that provide complementary signals during fine-tuning. LearnAlign (Li et al., 2025a) measures data influence through gradient alignment, providing a theoretically motivated approach to identifying influential samples. GRESO (Zheng et al., 2025) employs a probabilistic filtering strategy to exclude samples with historically zero variance, thereby reducing redundancy in training data. Other methods leverage either external models (Lu et al., 2023; Chen et al., 2023; Du et al., 2023; Liu et al., 2023) or internal proxy signals (Li et al., 2023a; Wu et al., 2023; Xia et al., 2024a; Yin et al., 2024; Liu et al., 2024a; Li et al., 2023b; Ivison et al., 2022) to rank training data, while they were originally designed for instruction fine-tuning. More recently, DEPO (Tang et al., 2025) introduces a unified data-efficient policy optimization pipeline that combines offline high-quality subset selection and online rollout filtering. By prioritizing diversity, influence, and difficulty, DEPO reduces data and computation requirements while maintaining strong reasoning performance.

Despite these advances, existing methods are typically outcome-oriented, relying on aggregate measures such as final answer correctness (e.g., LIM, GRESO) or loss reduction (e.g., LearnAlign). While such signals are useful, they overlook the process-oriented nature of multi-step reasoning, where the quality of intermediate steps can provide valuable insight into reasoning performance. As a result, current approaches may fail to identify the most informative reasoning samples. Our work addresses this gap by introducing a step-level uncertainty-aware selection framework that considers both step-level and outcome-level signals, providing a principled mechanism for identifying the reasoning samples most beneficial for RL post-training.

**Uncertainty Estimation of LLMs.** Uncertainty estimation in LLMs is gaining increasing attention as a mechanism for improving model calibration and mitigating hallucinations in text generation (Zhang et al., 2025; Vashurin et al., 2024). It has been adopted by popular scenarios in reasoning LLMs, including inference-time scaling (Xie et al., 2023), test-time adaptation (Zuo et al., 2025), and even post-training optimization Zhao et al. (2025) of reasoning models. Broadly, uncertainty estimation techniques can be grouped into two categories. The first are *logits-based methods*, which estimate uncertainty directly from token-level output distributions, such as predictive entropy or KL divergence from a uniform distribution (Fu et al., 2025; Ren et al., 2022; Duan et al., 2023; Darrin et al., 2022). The second are *verbalized uncertainty methods*, where models are prompted to explicitly articulate their confidence in natural language (Lin et al., 2022; Kadavath et al., 2022; Tian et al., 2023; Kapoor et al., 2024). A prominent representative of the latter is LLM *self-evaluation* (Kadavath et al., 2022), which leverages the model's own judgment as a more calibrated criterion to verify its predictions and guide reasoning trajectories.

Recent studies have begun extending both token-level and instance-level uncertainty to the finer granularity of *step-level uncertainty signals*, which are particularly suited for multi-step reasoning. For example, Xie et al. (2023) applied self-evaluation outcomes as criteria to calibrate stepwise generation, thereby addressing the challenges associated with complex or lengthy reasoning chains. Ye et al. (2025) introduced uncertainty-aware step-wise verification frameworks that explicitly model the reliability of intermediate steps to improve robustness in reasoning-heavy tasks.

Our work builds on this line of research by being the first to incorporate step-level uncertainty into data selection for reinforcement learning. Unlike prior outcome-oriented strategies, our method leverages both outcome-level and step-level uncertainties as a principled indicator of sample value during training, which in turn provides a novel connection between uncertainty estimation and data-efficient post-training of reasoning LLMs.

## 3 METHODOLOGY

### 3.1 UNCERTAINTY-BASED DATA SELECTION FRAMEWORK

As illustrated in Figure 1, our Star-DS framework integrates two complementary signals of reasoning uncertainty. Given an input question, the model first generates multiple reasoning rollouts, each decomposed into step-level units where the model self-evaluates its intermediate reasoning to produce

Figure 1: Overview of the proposed Star-DS framework. Given an input problem, the model generates multiple reasoning rollouts. We measure uncertainty from two complementary perspectives: *step-wise uncertainty*, which captures instability within intermediate reasoning steps, and *reward variability*, which quantifies divergence across final outcomes. Samples with higher combined uncertainty are prioritized for training, ensuring that the model focuses on resolving fragile reasoning paths and refining its decision-making process.

*step-wise uncertainty*. In parallel, we compute *reward variability* by measuring outcome divergence across different rollouts. These two complementary signals are then integrated into a composite uncertainty score, based on which the most uncertain samples are prioritized for training. This design ensures that the model allocates learning capacity to instances where reasoning is fragile or inconsistent, thereby enhancing both robustness and efficiency in multi-step reasoning tasks.

## 3.2 SELF-EVALUATED STEP-WISE UNCERTAINTY

To capture fine-grained reasoning uncertainty, we introduce a step-wise self-evaluation mechanism. Given an input sample $x$ from a dataset, the model generates multiple outputs, each referred to as a rollout $r$. Each rollout corresponds to a complete chain-of-thought (CoT), consisting of a sequence of intermediate reasoning steps. The generation of multi-step CoT is guided using a carefully designed prompt template (as shown in Appendix A), which encourages the model to produce detailed reasoning paths (Wang et al., 2022). Formally, the set of rollouts for a given input is denoted as:

$$R(x) = \{r^{(1)}, r^{(2)}, \ldots, r^{(k)}\}, \quad r^{(i)} \sim \pi_\theta(\cdot \mid x), \tag{1}$$

where $\pi_\theta$ represents the policy induced by the LLM with parameters $\theta$. Consider a rollout,

$$r = (s_1, s_2, \ldots, s_T), \tag{2}$$

where each $s_t$ denotes the $t^{th}$ reasoning step. Inspired by recent progress in uncertainty quantification (Ye et al., 2025; Zhang et al., 2025), we adopt a self-evaluation scenario: during each reasoning step, the model is prompted to evaluate its own reasoning by producing a binary judgment, either "correct" if the step is deemed valid or "incorrect" if it is potentially flawed. The evaluation context at step $t$ is defined as:

$$x_{\leq t} = (Q, s_{\leq t}, I_{\text{CoT}}), \tag{3}$$

where $Q$ represents the original question, $s_{\leq t}$ denotes the partial reasoning path up to step $t$, and $I_{\text{CoT}}$ is the chain-of-thought prompt provided for self-evaluation. Following (Kadavath et al., 2022), we design $I_{\text{CoT}}$ in the form of binary-choice questioning (see Appendix A) to better calibrate the model predictions. This self-assessment serves as a proxy for the model's internal uncertainty, allowing us to estimate where the reasoning process may be unstable or prone to error without relying on external verifiers or reward models.

The reasoning model then outputs a token-level probability distribution over the two possible decisions. We define the uncertainty of step $s_t$ as the probability assigned to the "incorrect" token:

$$U(s_t) = \pi_\theta(\text{"incorrect"} \mid x_{\leq t}). \tag{4}$$

The step-wise uncertainties are subsequently aggregated to form the rollout-level uncertainty:

$$U(r) = \frac{1}{T} \sum_{t=1}^{T} U(s_t). \tag{5}$$

Finally, the sample-level uncertainty is defined as the expectation over the rollout set:

$$U(x) = \mathbb{E}_{r^{(i)} \sim R(x)} \left[ U(r^{(i)}) \right]. \tag{6}$$

This formulation explicitly measures the uncertainty of the model across multiple reasoning steps. In contrast to outcome-only uncertainty measurements, it captures instabilities at intermediate steps and highlights samples that challenge the model's reasoning process across multiple reasoning stages.

### 3.3 REWARD VARIABILITY

While step-wise evaluation captures fine-grained uncertainties, it is also important to assess variability at the level of final outcomes (Razin et al., 2025; Gao et al., 2025). To this end, we introduce reward variability, which quantifies the degree of divergence in rollout-level results.

For each input prompt $x$, we generate $k$ rollouts and define the reward of the $i$-th rollout, $r_{\text{reward}}^{(i)}$, as an indicator of final correctness. Following the setting of popular reinforcement learning with verifiable rewards (RLVR), $r_{\text{reward}}^{(i)} = 1$ if the predicted answer matches the ground truth, and $r_{\text{reward}}^{(i)} = 0$ otherwise (Gao et al., 2024; Lambert et al., 2024). The reward variability is then computed as the standard deviation of all the rollout rewards:

$$\sigma_{\text{reward}}(x) = \sqrt{\frac{1}{k} \sum_{i=1}^{k} \left( r_{\text{reward}}^{(i)} - \bar{r}_{\text{reward}} \right)^2}, \tag{7}$$

where $\bar{r}_{\text{reward}} = \frac{1}{k} \sum_{i=1}^{k} r_{\text{reward}}^{(i)}$ denotes the mean reward across all rollouts.

High reward variance indicates that the model's policy has not yet converged for the given sample, as different reasoning paths yield inconsistent outcomes—some trajectories succeed while others fail (Foster et al., 2025; Rutherford et al., 2024). Samples exhibiting such variability are particularly informative for training, as they highlight regions where the model remains uncertain about its overall strategy and can benefit from additional supervision or learning signals.

### 3.4 COMBINED UNCERTAINTY SCORING AND DATA SELECTION

To leverage both step-wise reasoning uncertainty and outcome-level variability, we combine the two uncertainty metrics into a single composite score. Specifically, for a given sample $x$, we compute the step-wise uncertainty $U(x)$ and the reward standard deviation $\sigma_{\text{reward}}(x)$ as described in the previous subsections. Since the two metrics may be measured on different scales, we scale them separately to the range $[0, 1]$. Denote the normalized values as $\tilde{U}(x)$ and $\tilde{\sigma}_{\text{reward}}(x)$, respectively.

The final uncertainty score is then obtained by summing the normalized components:

$$U_{\text{final}}(x) = \tilde{U}(x) + \tilde{\sigma}_{\text{reward}}(x). \tag{8}$$

This formulation ensures that both intermediate reasoning instability and final outcome divergence contribute in a balanced manner to the overall uncertainty score, capturing complementary aspects of model uncertainty.

Once the composite scores are computed for all samples in the dataset, we perform top-$k$ selection to identify the most informative examples. Formally, given a dataset $\mathcal{D}$ and a selection budget $k$, the samples are selected according to

$$\mathcal{D}_{\text{selected}} = \text{TopK} \left( \{ U_{\text{final}}(x) \mid x \in \mathcal{D} \}, k \right), \tag{9}$$

where $\text{TopK}(\cdot, k)$ returns the $k$ samples with the highest uncertainty scores. These selected samples are expected to provide the strongest learning signals, as they represent instances where the model exhibits either unstable reasoning, divergent outcomes, or both. Consequently, training on $\mathcal{D}_{\text{selected}}$ can improve model robustness and efficiency in reasoning tasks.

## 4 EXPERIMENTS

### 4.1 EXPERIMENTAL SETUP

**Base Models & Data Selection.** We adopt Qwen2.5-Math-1.5B (Yang et al., 2024; 2025) as the primary base reasoning model for reasoning data selection. We additionally verify the effectiveness of DeepSeek-R1-Distill-Qwen-1.5B (Guo et al., 2025). We perform data selection on a candidate pool that consists of two datasets: MATH (Hendrycks et al., 2021) training set with 7,500 instances, and GSM8K (Cobbe et al., 2021) training set with 7473 instances. For data selection, we generate 8 reasoning rollouts with a temperature of 0.6 for each candidate instance, and compute both stepwise uncertainty and reward variability. The scores are then aggregated to identify the top-ranked samples, which are used for downstream fine-tuning. We provide detailed examples of selected data in Appendix C.

**Training Setup.** The proposed framework is conducted with the GRPO (Shao et al., 2024) algorithm following the verl pipeline (Sheng et al., 2025). For fine-tuning on the selected subsets, we maintain consistent hyperparameter settings across all experiments: batch size is 64, learning rate is $1 \times 10^{-6}$, the maximum prompt length is 1024 tokens, and the maximum response length is 3072 tokens. KL divergence regularization is applied with a coefficient of 0.001, and the clipping parameter is set to 0.2. All models are trained for 200 epochs to ensure fair comparison.

**Evaluation Protocol.** To verify the effectiveness of the selected reasoning data, we fine-tune the base model on the selected subsets and evaluate under the official Qwen2.5-Math evaluation pipeline (Yang et al., 2024) across six benchmark datasets: MATH500 (Hendrycks et al., 2021; Lightman et al., 2023), AIME 2024 (AIM), AMC 2023 (AMC), Minerva Math (Lewkowycz et al., 2022), OlympiadBench (He et al., 2024), and AIME 2025 (AIM). For AIME 2024, AIME 2025, and AMC 2023, which consist of only 30 or 40 questions each, we repeat the test set 8 times to ensure evaluation stability, using a rollout temperature of 0.6 and reporting the average pass@1 (avg@8) performance. For the remaining three benchmarks, we set the temperature to 0 and evaluate using the standard protocol. Further experimental details are reported in Appendix B.

**Baselines.** We compare Star-DS against the baseline strategies below: (1) **Random Sampling**, which selects samples uniformly at random and serves as a control group; (2) **Learning Impact Measurement (LIM)** (Li et al., 2025b), which prioritizes samples whose learning patterns complement the model's overall performance trajectory; (3) **PPL** (Laurençon et al., 2022), which selects samples with highest perplexity under the pretrained model; (4) **Instruction-Following Difficulty (IFD)** (Li et al., 2023a), which measures the challenge of each instructional sample for the model; (5) **Token Length** (Xia et al., 2024b), which ranks samples based on their token length. These baselines cover both simple heuristics and more principled, model-aware approaches for data selection.

### 4.2 MAIN RESULTS

Table 1 presents the comparison between Star-DS and baselines. Overall, Star-DS consistently outperforms all baselines across nearly all benchmarks. On AIME 2024, AMC 2023, and MATH500, it achieves pass@1 scores of 16.2, 54.4, and 73.4, outperforming the best baseline by margins of 0.4, 1.9, and 0.8 points, respectively. These performance gains indicate that selecting samples based on step-level uncertainty and reward variability effectively enhances multi-step reasoning capabilities.

Compared to random and perplexity-based selection, Star-DS shows clear advantages, suggesting that simple heuristics are insufficient to capture the intricacies of reasoning difficulty. While LIM and IFD incorporate learning dynamics or instruction-following difficulty, these signals might be less sensitive to the subtle variations that influence multi-step reasoning performance, thus lagging behind Star-DS. Notably, Star-DS even rivals or surpasses training on the full MATH dataset (e.g., 35.3 vs. 32.6 on OlympiadBench), demonstrating that careful subset selection not only improves efficiency but also enhances reasoning robustness.

**Computational Efficiency.** From an efficiency perspective, training the selected 1,000-sample subset for a single epoch takes roughly 90 seconds, whereas training the full 7,500-sample dataset for one epoch takes about 240 seconds (on the same hardware). Even when including the overhead for metric evaluation, Star-DS remains efficient. Based on our experimental measurements, completing a full

Table 1: Comparison of data selection methods on six mathematical reasoning benchmarks with Qwen2.5-Math-1.5B. All models are trained with GRPO for 200 epochs on 1,000-example subsets of MATH selected by each method unless noted. "NA" denotes the base model without additional training, and "MATH-FULL" uses the full 7,500-example MATH set. Best results (pass@1) are **bolded**, and second-best are underlined.

| Selection Method | AIME24 | AIME25 | AMC23 | MATH500 | Minerva | Olympiad | Average |
|---|---|---|---|---|---|---|---|
| NA | 6.7 | 4.6 | 33.1 | 40.2 | 9.6 | 22.7 | 19.5 |
| MATH-FULL | 15.0 | **8.8** | 51.6 | 72.6 | **28.7** | 32.6 | 34.9 |
| Random | 15.4 | 7.5 | 52.2 | 72.2 | 27.5 | 35.0 | 35.0 |
| LIM | 15.4 | 8.3 | 52.5 | 71.4 | 26.5 | 34.1 | 34.7 |
| PPL | 15.8 | 7.1 | 50.6 | 71.8 | 27.6 | 34.1 | 34.5 |
| IFD | 15.8 | 7.1 | 52.2 | 72.6 | 28.3 | 34.8 | 35.1 |
| Star-DS | **16.2** | **8.8** | **54.4** | **73.4** | 28.3 | **35.3** | **36.1** |

Table 2: Ablation study (pass@1) on six benchmarks with Qwen2.5-Math-1.5B. All models are trained with GRPO for 200 epochs on 1,000 selected samples of MATH. "Stepwise uncertainty" and "Reward variability" correspond to using each signal individually, while "Star-DS" combines both. Best results are **bolded**, and second-best are underlined.

| Method | AIME24 | AIME25 | AMC23 | MATH500 | Minerva | Olympiad | Average |
|---|---|---|---|---|---|---|---|
| Stepwise Uncertainty | 14.6 | 7.9 | **54.5** | 73.0 | 27.4 | 34.5 | 35.3 |
| Reward Variability | 14.6 | 8.3 | 53.8 | 72.6 | **28.3** | **35.9** | 35.6 |
| Star-DS | **16.2** | **8.8** | 54.4 | **73.4** | **28.3** | 35.3 | **36.1** |

estimation of reward variability roughly corresponds to training the entire dataset for one epoch, while calculating stepwise uncertainty is equivalent to approximately five epochs over the full dataset. These measurements highlight that, despite the additional evaluation overhead, Star-DS significantly reduces overall computation time compared to training on the full dataset.

## 4.3 ABLATION STUDY

To better understand the contribution of each component in Star-DS, we conduct an ablation study with a fixed budget of 1,000 selected samples. Specifically, we compare three variants: using only **stepwise uncertainty**, using only **reward variability**, and the proposed **Star-DS** method that combines both signals. All models are trained under the same experimental settings as the main experiment.

Table 2 presents the results. We observe that even when using a single scoring criterion, the selected data already achieves competitive improvements, outperforming most baseline methods. Specifically, reasoning data selected by stepwise uncertainty only achieves the best performance on AMC23 (54.5), while that selected by reward variability only leads on OlympiadBench (35.9) and Minerva (28.3). Data selected by Star-DS with both criteria consistently achieves the best or near-best results across almost all benchmarks, including the highest scores on AIME24 (16.2), AIME25 (8.8), and MATH500 (73.4). These findings highlight the complementary nature of the two metrics and demonstrate that combining them produces a more balanced and robust selection strategy.

## 4.4 EFFECT OF SELECTED DATA SIZE

We further investigate the impact of the number of selected training samples on method performance. We fix the training epochs to 200 and vary the selection subset size among 100, 500, and 1,000 examples for each selection method. Table 3 reports the average pass@1 score across six mathematical reasoning benchmarks.

Overall, our Star-DS consistently outperforms baseline selection strategies at all subset sizes, demonstrating both efficiency and robustness. Specifically, even with only 100 selected examples, Star-DS achieves an average pass@1 of 34.85, nearly doubling the base model (19.48) and already surpassing some larger subsets of baseline methods. As the selected data size increases to 1,000 (13.4% of full data), performance rises to 36.07, exceeding the result of training on the full MATH dataset (34.88),

Table 4: Performance (pass@1) on six mathematical reasoning benchmarks using GSM8K as the training pool. All models are trained with GRPO for 200 epochs on 1,000-example subsets selected by each method unless noted. "NA" denotes the base model without additional training, and "GSM8K-FULL" uses the entire GSM8K training set for comparison. Best results are **bolded**, and second-best are underlined.

| Selection Method | AIME24 | AIME25 | AMC23 | MATH500 | Minerva | Olympiad | Average |
|---|---|---|---|---|---|---|---|
| NA | 6.7 | 4.6 | 33.1 | 40.2 | 9.6 | 22.7 | 19.5 |
| GSM8K-FULL | 14.2 | 8.8 | 51.6 | 72.8 | 25.4 | 34.8 | 34.6 |
| Random | 14.6 | 8.8 | 51.5 | 71.4 | 25.7 | 34.2 | 34.4 |
| Token Length | 12.5 | 7.9 | 50.9 | **73.8** | 18.4 | 34.3 | 32.9 |
| PPL | 14.2 | 8.3 | 51.9 | 70.0 | 25.0 | 33.0 | 33.7 |
| IFD | 14.2 | 9.0 | 50.6 | 73.2 | **27.3** | 33.8 | 34.7 |
| Star-DS | **15.0** | **9.2** | **52.2** | **73.8** | 26.5 | **35.3** | **35.3** |

highlighting that carefully curated subsets can be more informative than the full training set. These results quantitatively demonstrate that Star-DS not only selects high-quality training data efficiently but also scales effectively, providing superior multi-step reasoning performance with fewer examples. For completeness, the benchmark-wise results for different subset sizes are reported in Appendix B.4.

## 4.5 OTHER DATASET AND MODEL

To evaluate the generalizability of Star-DS, we conduct two additional experiments: (1) using GSM8K as the data pool for dataset-level validation with Qwen2.5-Math-1.5B, and (2) using DeepSeek-R1-Distill-Qwen-1.5B as the base model for model-level validation. In both cases, selected subsets are trained for 200 epochs and evaluated on six mathematical reasoning benchmarks, with fixed selection budgets of 1,000 and 100 examples, respectively.

On GSM8K (see Table 4), Star-DS consistently achieves the best or near-best performance across benchmarks. Notably, carefully selected subsets can match or surpass training on the full GSM8K dataset (e.g., 35.3 vs. 34.8 on OlympiadBench), demonstrating effective identification of high-value reasoning samples across domains.

For DeepSeek-R1-Distill-Qwen-1.5B (see Table 5), Star-DS consistently achieves the highest performance, except when compared to training on the full

Table 3: Average pass@1 performance across six benchmarks under three data selection sizes. All models are trained with GRPO for 200 epochs. "NA" denotes the base Qwen2.5-Math-1.5B model without additional training, and "MATH-FULL" uses the full 7,500-example MATH set. Best results are **bolded**.

| Selection Method | Selection Size | | |
| | 100 | 500 | 1,000 |
|---|---|---|---|
| Random | 33.73 | 34.00 | 34.97 |
| LIM | 34.25 | 33.83 | 34.70 |
| PPL | 33.02 | 33.45 | 34.50 |
| IFD | 34.30 | 34.15 | 35.13 |
| Star-DS | **34.85** | **34.77** | **36.07** |
| NA | | 19.48 | |
| MATH-FULL | | 34.88 | |

dataset. In contrast, the lowest-ranked samples identified by Star-DS (Star-DS_Bottom) perform significantly worse than random selection. It demonstrates that Star-DS effectively distinguishes high-value from low-value reasoning samples, confirming that Star-DS is meaningful and reliable for different base models.

## 5 ANALYSIS

Figure 2 illustrates the training dynamics on the MATH dataset for four different data selection strategies: (1) training on the full dataset, (2) selecting samples based solely on stepwise uncertainty, (3) selecting samples based solely on reward variability, and (4) selecting samples using the combination of stepwise uncertainty and reward variability (the proposed Star-DS). The figure highlights both reward progression and response length trends over 200 epochs.

As shown by the **reward curve** (left), the stepwise uncertainty strategy starts from a relatively low average reward of 0.4 and gradually increases to 0.55 after 200 epochs, reflecting slower initial

Table 5: Performance (pass@1) of DeepSeek-R1-Distill-Qwen-1.5B trained on 100-example subsets selected by each method for 200 epochs. "NA" denotes the base model without additional training, and "MATH-FULL" uses the entire training set. "Star-DS_Bottom" consists of samples ranked lowest by Star-DS. Best results are **bolded**, and second-best are underlined.

| Selection Method | AIME24 | AIME25 | AMC23 | MATH500 | Minerva | Olympiad | Average |
|---|---|---|---|---|---|---|---|
| NA | 12.1 | 10.8 | 45.3 | 63.6 | 15.8 | 24.6 | 28.7 |
| MATH-FULL | **20.0** | **20.0** | **65.3** | **81.0** | **28.7** | **41.0** | **42.7** |
| Random | 18.3 | 18.8 | 63.4 | 77.0 | 24.6 | 40.1 | 40.4 |
| Star-DS_Bottom | 17.4 | 18.3 | 61.6 | 75.8 | 23.8 | 39.0 | 39.3 |
| Star-DS | 18.3 | 19.2 | 64.4 | 79.0 | 25.3 | 40.6 | 41.1 |

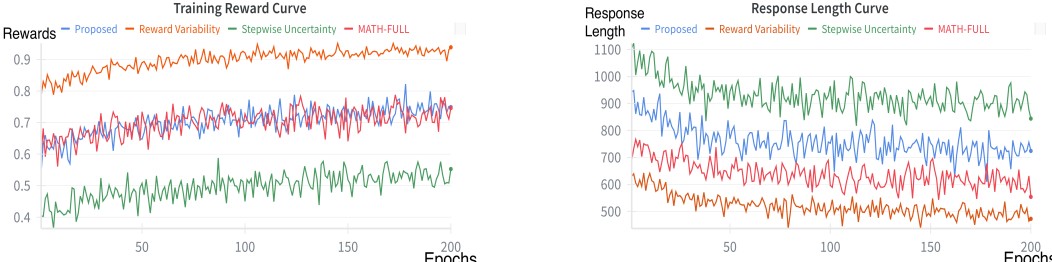

Figure 2: **Training dynamics of different selection strategies. Left:** Reward curves of four training settings on the MATH dataset: training on the full set, stepwise uncertainty only, reward variability only, and their combination Star-DS (proposed). **Right:** Response length curves under the same settings.

learning but steady improvement. In contrast, the reward variability strategy begins with a much higher reward of 0.8 and continues to rise to approximately 0.9, indicating that the selected samples are initially more informative for maximizing immediate outcomes. The combined method exhibits a balanced trajectory, starting at 0.6 and reaching 0.75, closely mirroring the reward curve observed when training on the full dataset. This demonstrates that integrating both uncertainty signals captures complementary aspects of the training data, achieving stable and effective reward improvement.

On the **response length curve** (right), stepwise uncertainty produces the longest responses, beginning around 1,100 tokens and decreasing to 850 tokens after 200 epochs. Reward variability generates much shorter responses, decreasing from 650 to roughly 500 tokens. The combined strategy yields an intermediate effect, with response lengths starting from 950 and descending to about 750 tokens, exhibiting a similar downward trend but remaining consistently above the full dataset curve, which drops from approximately 750 to 550 tokens. These observations suggest that Star-DS produces outputs of moderate length, striking a balance between the verbosity of stepwise uncertainty and the brevity of reward variability.

## 6 CONCLUSION

In this paper, we make the first attempt at reasoning data selection for LLM multi-step reasoning, introducing a step-level uncertainty-aware framework Star-DS. Star-DS integrates both step-wise self-evaluation uncertainty and reward variability to more effectively capture the intrinsic challenges of multi-step reasoning. Experiments across diverse benchmarks and reasoning models demonstrate that Star-DS consistently identifies high-value data, preserves reasoning performance, and substantially reduces data and computational requirements. In future work, we intend to extend this framework to online data selection, enabling dynamic identification of valuable training instances during reinforcement learning.

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

APPENDIX

In this appendix, Section A presents the full prompt templates used for rollout generation and step-wise self-evaluation in our framework. Section B describes the detailed experimental setup, including the GRPO algorithm, the training datasets, and the evaluation benchmarks. Extended results across different data subset sizes are also reported. Section C provides concrete examples of selected data based on our proposed scoring strategies, including stepwise uncertainty, reward variability, and their combination. Section D clarifies the scope of LLM usage within our framework. Finally, our code is provided in the supplementary material and can also be accessed via `https://anonymous.4open.science/r/Star-DS-5961`.

## A    PROMPT TEMPLATES FOR ROLLOUT GENERATION AND SELF-EVALUATION

For reproducibility, we provide the exact prompt templates used in our framework. The first template is designed for **rollout generation**, where the model is instructed to produce detailed step-by-step chain-of-thought solutions to mathematical problems. To better guide the model's reasoning format, this template includes several worked examples before the target question. The second template is used for **step-wise self-evaluation**, where the model assesses the correctness of each reasoning step in a binary fashion ("Correct" or "Incorrect") given a specific math problem and the reasoning steps generated so far. We reproduce the full text of both prompts below.

---

**Prompt Template for Rollout Generation**

Q: What is the value of $2^{0^{1^9}} + (2^0)^{1^9}$?
A:

Step 1: Evaluate the first term $2^{\left(0^{\left(1^9\right)}\right)}$ - First, simplify the exponent's base: $1^9 = 1$. - Then, calculate $0^1 = 0$. - Finally, compute $2^0 = 1$.

Step 2: Evaluate the second term $\left(\left(2^0\right)^1\right)^9$ - Simplify inside the parentheses: $2^0 = 1$. - Then, $(1)^1 = 1$. - Raise 1 to the power of 9: $1^9 = 1$.

Step 3: Add the results of the two terms - $1 + 1 = 2$. Thus, the value of the expression is $\boxed{2}$.

Q: Chandler wants to buy a 500 dollar mountain bike. For his birthday, his grandparents send him 50 dollars, his aunt sends him 35 dollars and his cousin gives him 15 dollars. He earns 16 dollars per week for his paper route. He will use all of his birthday money and all of the money he earns from his paper route. In how many weeks will he be able to buy the mountain bike?
A:
Step 1. **Identify the total amount of money Chandler needs:** Chandler wants to buy a mountain bike that costs 500.

Step 2. **Calculate the total amount of birthday money Chandler received:** - From his grandparents: 50 - From his aunt: 35 - From his cousin: 15 - Total birthday money: $50 + 35 + 15 = 100$

Step 3. **Set up an equation for the total money Chandler will have after $x$ weeks of saving his earnings from the paper route:** - He earns 16 per week from his paper route. - Total money after $x$ weeks: Birthday money + Earnings from paper route = $100 + 16x$

Step 4. **Formulate the equation based on the cost of the mountain bike:** - The total money should equal the cost of the bike: $100 + 16x = 500$

Step 5. **Solve the equation for $x$:** - Subtract 100 from both sides of the equation: $16x = 500 - 100$ - Simplify the right side: $16x = 400$ - Divide both sides by 16: $x = \frac{400}{16}$ - Simplify the division: $x = 25$

Step 6. **Conclusion:** Chandler will be able to buy the mountain bike after saving for 25 weeks. The answer is $\boxed{25}$.

---

Q: Six rectangles each with a common base width of 2 have lengths of $1, 4, 9, 16, 25$, and $36$. What is the sum of the areas of the six rectangles?

A:

Step 1. **Identify the dimensions of each rectangle**: Each rectangle has a common base width of 2. The lengths of the rectangles are given as $1, 4, 9, 16, 25$, and $36$.

Step 2. **Calculate the area of each rectangle**: The area of a rectangle is calculated by multiplying its length by its width. Therefore, the area of each rectangle can be calculated as follows: - For the rectangle with length 1: Area = $2 \times 1 = 2$ - For the rectangle with length 4: Area = $2 \times 4 = 8$ - For the rectangle with length 9: Area = $2 \times 9 = 18$ - For the rectangle with length 16: Area = $2 \times 16 = 32$ - For the rectangle with length 25: Area = $2 \times 25 = 50$ - For the rectangle with length 36: Area = $2 \times 36 = 72$

Step 3. **Sum the areas of all rectangles**: Add the areas calculated in the previous step:

$$\text{Total Area} = 2 + 8 + 18 + 32 + 50 + 72$$

Step 4. **Simplify the sum**: Add the numbers to find the total area:

$$\text{Total Area} = 2 + 8 + 18 + 32 + 50 + 72 = 182$$

Therefore, the sum of the areas of the six rectangles is $\boxed{182}$.

=== End of Example ===

Instruction for the next problem:

When solving the next question, please explicitly follow the same format as the above example:
- MUST Use Step 1:, Step 2:, Step 3:, etc.
- MUST NOT generate new questions, examples, verification code, or explanations beyond this problem.
- MUST end your answer with the final value inside $\square$.

Q: {question}
A:

---

## Prompt Template for Self-Evaluation

You are an expert math reasoning evaluator.
Your ONLY task is to evaluate the correctness of the CURRENT reasoning step,
given the previous reasoning steps.
Do NOT try to solve the original problem or guess future steps.

Here is a math problem:
{question}

Here is the reasoning so far:
{previous_steps}

Now, consider the next reasoning step:
{current_step}

Please carefully evaluate whether THIS step is correct given the reasoning so far.
Choose ONE of the following options as your FINAL answer:
# (A) Correct
# (B) Incorrect

IMPORTANT: Your final output must be a single character: either "A" or "B".

## B   DETAILED EXPERIMENTAL SETUP AND ADDITIONAL RESULTS

In this section, we provide a comprehensive overview of our experimental setup and additional experimental results. We first describe the Group Relative Policy Optimization (GRPO) algorithm used for fine-tuning large language models (see Section B.1). Next, we detail the training datasets employed, including MATH and GSM8K (see Section B.2). We then summarize the evaluation benchmarks used to assess model performance, including MATH500, AIME 2024/2025, AMC 2023, Minerva Math, and OlympiadBench (see Section B.3). Finally, we present the detailed results for selected data subsets of varying sizes (see Section B.4).

### B.1   DETAILS OF GROUP RELATIVE POLICY OPTIMIZATION (GRPO) ALGORITHM

We adopt the Group Relative Policy Optimization (GRPO) algorithm (Shao et al., 2024) for fine-tuning reasoning models. GRPO is designed for multi-step reasoning tasks and emphasizes group-relative performance: each rollout is evaluated relative to a group of rollouts, encouraging the model to favor consistently effective reasoning chains.

Formally, let $\pi_\theta$ denote the policy of the model with parameters $\theta$. For a given input $q$ and its associated group of $G$ rollouts $\{o_i\}_{i=1}^G$, the GRPO objective is defined as:

$$J_{\text{GRPO}}(\theta) = \mathbb{E}_{(q,a)\sim P_q, \{o_i\}_{i=1}^G \sim \pi_\theta^{\text{old}}(o|q)} \left[ \frac{1}{G} \sum_{i=1}^G \frac{1}{|o_i|} \sum_{t=1}^{|o_i|} \min\left(r_{i,t}\hat{A}_{i,t}, \text{clip}(r_{i,t}, 1-\epsilon, 1+\epsilon)\hat{A}_{i,t}\right) \right.$$

$$\left. - \beta\, \text{KL}[\pi_\theta \,||\, \pi_{\text{ref}}] \right], \tag{10}$$

where $r_{i,t} = \frac{\pi_\theta(o_{i,t}|q, o_{i,<t})}{\pi_\theta^{\text{old}}(o_{i,t}|q, o_{i,<t})}$, and the relative advantage $\hat{A}_{i,t}$ is computed as

$$\hat{A}_{i,t} = \frac{r_i - \text{mean}(\{r_j\}_{j=1}^G)}{\text{std}(\{r_j\}_{j=1}^G)}. \tag{11}$$

Here, $\pi_{\text{ref}}$ represents the reference (pre-trained) model, while $\epsilon$ and $\beta$ are hyperparameters controlling the clipping range and KL-regularization weight, respectively. This formulation ensures that updates favor rollouts performing better than the group average while maintaining proximity to the pre-trained policy.

In our experiments, GRPO is applied to fine-tune selected subsets of reasoning data. Each candidate input generates multiple rollouts, and the group-relative advantage guides the policy to prioritize stable, high-quality reasoning chains, while the KL term stabilizes updates and prevents catastrophic deviation from the pre-trained model.

### B.2   DETAILS OF TRAINING DATASETS

**MATH.** The MATH dataset (Hendrycks et al., 2021) contains 12,500 challenging mathematics problems sourced from high school-level competitions, designed to assess advanced problem-solving skills in machine learning models. Problems span topics including Prealgebra, Algebra, Number Theory, Counting and Probability, Geometry, Intermediate Algebra, and Precalculus. Each problem is assigned a difficulty level from 1 to 5 and includes detailed step-by-step solutions. For training purposes, we use the designated training split of 7,500 problems (60% of the full dataset).

**GSM8K.** GSM8K (Cobbe et al., 2021) is a collection of 8,500 grade school mathematics word problems emphasizing multi-step reasoning. Each problem generally requires 2–8 reasoning steps and can be solved with basic arithmetic operations (+, -, $\times$, $\div$). Solutions are provided in natural language, encouraging models to generate coherent reasoning chains in addition to the final answer. The dataset is linguistically diverse and crafted to be solvable by a bright middle school student, making it a useful complement to MATH for reasoning data selection.

## B.3 DETAILS OF EVALUATION DATASETS

**MATH500.** MATH500 (Hendrycks et al., 2021) is a curated subset of 500 problems drawn from the MATH test split. This smaller benchmark is used to facilitate efficient and reproducible evaluation while maintaining problem difficulty diversity.

**AIME 2024/2025.** These datasets consist of 30 problems each from the 2024 and 2025 American Invitational Mathematics Examination (AIME) I and II. They provide a focused evaluation of multi-step reasoning on challenging competition-level problems (AIM).

**AMC 2023.** The AMC 2023 benchmark contains 40 problems selected from the AMC 12A and 12B competitions for U.S. high school students. Topics include arithmetic, algebra, combinatorics, geometry, number theory, and probability, with all problems solvable without calculus (AMC).

**Minerva Math.** Minerva Math (Lewkowycz et al., 2022) is a set of 272 undergraduate STEM problems derived from MIT OpenCourseWare. The dataset emphasizes multi-step scientific reasoning across courses such as solid-state chemistry, information theory, differential equations, and special relativity. Problems are self-contained, and solutions are either numeric or symbolic.

**OlympiadBench.** OlympiadBench (He et al., 2024) is a large-scale benchmark for advanced mathematical and physical reasoning. The subset used for evaluation comprises 675 text-only, open-ended problems in English sourced from international math competitions, with expert-annotated step-by-step solutions.

## B.4 DETAILED RESULTS ACROSS DIFFERENT SUBSET SIZES

In this section, we present the detailed benchmark-wise performance corresponding to the averaged outcomes reported in Section 4.4. Specifically, we provide results for three different subset sizes—100, 500, and 1,000 selected training samples—evaluated across six mathematical reasoning benchmarks. For each setting, we include the results of multiple baseline selection strategies as well as Star-DS, allowing a fine-grained comparison on individual benchmarks. These tables complement the main results by illustrating how each data selection method performs under varying amounts of training data. The complete results are provided in Tables 6–8.

Table 6: Results (pass@1) with 100 selected training samples.

| Selection Method | AIME24 | AIME25 | AMC23 | MATH500 | Minerva | Olympiad |
|---|---|---|---|---|---|---|
| Full | 15.0 | 8.8 | 51.6 | 72.6 | 28.7 | 32.6 |
| Random | 15.4 | 7.1 | 51.2 | 71.0 | 25.0 | 32.7 |
| IFD | 15.8 | 6.7 | 51.6 | 72.8 | 25.7 | 33.2 |
| LIM | 15.0 | 8.3 | 53.1 | 71.6 | 23.5 | 34.0 |
| PPL | 13.8 | 6.7 | 51.6 | 69.4 | 25.0 | 31.6 |
| Star-DS | 15.8 | 8.3 | 53.2 | 71.2 | 26.8 | 33.8 |

Table 7: Results (pass@1) with 500 selected training samples.

| Selection Method | AIME24 | AIME25 | AMC23 | MATH500 | Minerva | Olympiad |
|---|---|---|---|---|---|---|
| Full | 15.0 | 8.8 | 51.6 | 72.6 | 28.7 | 32.6 |
| Random | 14.8 | 7.9 | 50.9 | 70.4 | 26.1 | 33.9 |
| IFD | 14.5 | 7.1 | 51.6 | 72.4 | 26.1 | 33.2 |
| LIM | 13.8 | 5.8 | 52.8 | 70.6 | 27.9 | 32.1 |
| PPL | 12.9 | 7.5 | 49.7 | 70.6 | 26.8 | 33.2 |
| Star-DS | 14.6 | 7.1 | 52.5 | 72.0 | 27.6 | 34.8 |

Table 8: Results (pass@1) with 1,000 selected training samples.

| Selection Method | AIME24 | AIME25 | AMC23 | MATH500 | Minerva | Olympiad |
|---|---|---|---|---|---|---|
| Full | 15.0 | 8.8 | 51.6 | 72.6 | 28.7 | 32.6 |
| Random | 15.4 | 7.5 | 52.2 | 72.2 | 27.5 | 35.0 |
| IFD | 15.8 | 7.1 | 52.2 | 72.6 | 28.3 | 34.8 |
| LIM | 15.4 | 8.3 | 52.5 | 71.4 | 26.5 | 34.1 |
| PPL | 15.8 | 7.1 | 50.6 | 71.8 | 27.6 | 34.1 |
| Star-DS | 16.2 | 8.8 | 54.4 | 73.4 | 28.3 | 35.3 |

## C   EXAMPLE DETAILS

In this section, we present detailed examples of data selected based on our data scoring methods. For each scoring criterion—stepwise uncertainty, reward variability, and their combination—we show the top-5 and bottom-5 data samples according to the respective scores. Each example includes the full prompt, the ground-truth answer, and the computed score, providing an illustrative view of how different scoring strategies prioritize data. The corresponding examples are listed in Table 9, Table 10, and Table 11.

## D   THE USE OF LARGE LANGUAGE MODELS

In this work, we used large language models (LLMs) solely as a general-purpose tool to assist in polishing the writing and improving the clarity of the manuscript. The LLM was not involved in the formulation of research ideas, the design of experiments, or the analysis of results. All technical content, experimental design, data processing, and interpretations presented in this paper were independently developed by the authors.

We acknowledge that while the LLM contributed to text refinement, the authors take full responsibility for all content, including any text generated with LLM assistance. No LLM was listed as an author or contributor beyond its role in language polishing.

Table 9: Details of Top-5 and Bottom-5 examples ranked by stepwise uncertainty score.

| Rank | Index | Prompt | Ground Truth | Score |
|---|---|---|---|---|
| **Top-5 Examples** | | | | |
| Top-1 | 3087 | Let $S$ be a square of side length 1. Two points are chosen independently at random on the sides of $S$. The probability that the straight-line distance between the points is at least $\frac{1}{2}$ is $\frac{a - b\pi}{c}$, where $a$, $b$, and $c$ are positive integers with $\gcd(a, b, c) = 1$. What is $a + b + c$? Let's think step by step and output the final answer within \boxed. | 59 | 1.00 |
| Top-2 | 3923 | Let $a, b, c, d$ be real numbers such that $a + b + c + d = 6$, $a^2 + b^2 + c^2 + d^2 = 12$. Let $m$ and $M$ denote minimum and maximum values of $4(a^3 + b^3 + c^3 + d^3) - (a^4 + b^4 + c^4 + d^4)$, respectively. Find $m + M$. Let's think step by step and output the final answer within \boxed. | 84 | 1.00 |
| Top-3 | 4304 | The sequences of positive integers $1, a_2, a_3, \ldots$ and $1, b_2, b_3, \ldots$ are an increasing arithmetic sequence and an increasing geometric sequence, respectively. Let $c_n = a_n + b_n$. There is an integer $k$ such that $c_{k-1} = 100$ and $c_{k+1} = 1000$. Find $c_k$. Let's think step by step and output the final answer within \boxed. | 262 | 1.00 |
| Top-4 | 4601 | $\zeta_1, \zeta_2$, and $\zeta_3$ are complex numbers such that $\zeta_1 + \zeta_2 + \zeta_3 = 1, \zeta_1^2 + \zeta_2^2 + \zeta_3^2 = 3, \zeta_1^3 + \zeta_2^3 + \zeta_3^3 = 7$. Compute $\zeta_1^7 + \zeta_2^7 + \zeta_3^7$. Let's think step by step and output the final answer within \boxed. | 71 | 1.00 |
| Top-5 | 7222 | Find the equation of the plane passing through the point $(0, 7, -7)$ and containing the line $\frac{x+1}{-3} = \frac{y-3}{2} = \frac{z+2}{1}$. Enter your answer in the form $Ax + By + Cz + D = 0$, where $A, B, C, D$ are integers such that $A > 0$ and $\gcd(|A|, |B|, |C|, |D|) = 1$. Let's think step by step and output the final answer within \boxed. | $x + y + z = 0$ | 1.00 |
| **Bottom-5 Examples** | | | | |
| Bottom-1 | 2924 | In tetrahedron $ABCD$, edge $AB$ has length 3 cm. The area of face $ABC$ is 15cm$^2$ and the area of face $ABD$ is 12 cm$^2$. These two faces meet each other at a $30°$ angle. Find the volume of the tetrahedron in cm$^3$. Let's think step by step and output the final answer within \boxed. | 20 | 0.00 |
| Bottom-2 | 4009 | An integer-valued function $f$ is called tenuous if $f(x) + f(y) > y^2$ for all positive integers $x$ and $y$. Let $g$ be a tenuous function such that $g(1) + g(2) + \cdots + g(20)$ is as small as possible. Compute the minimum possible value for $g(14)$. Let's think step by step and output the final answer within \boxed. | 136 | 0.00 |
| Bottom-3 | 4523 | Is the function $f(x) = \lfloor x \rfloor + \frac{1}{2}$ even, odd, or neither? Enter "odd", "even", or "neither". Let's think step by step and output the final answer within \boxed. | neither | 0.00 |
| Bottom-4 | 4662 | The complex number $z$ traces a circle centered at the origin with radius 2. Then $z + \frac{1}{z}$ traces a: (A) circle (B) parabola (C) ellipse (D) hyperbola. Enter the letter of the correct option. Let's think step by step and output the final answer within \boxed. | C | 0.00 |
| Bottom-5 | 5059 | Given that $8^{-1} \equiv 85 \pmod{97}$, find $64^{-1} \pmod{97}$, as a residue modulo 97. (Give an answer between 0 and 96, inclusive.) Let's think step by step and output the final answer within \boxed. | 47 | 0.00 |

Table 10: Details of Top-5 and Bottom-5 examples ranked by reward variability score.

| Rank | Index | Prompt | Ground Truth | Score |
|---|---|---|---|---|
| **Top-5 Examples** | | | | |
| Top-1 | 4 | Sam is hired for a 20-day period. On days that he works, he earns \$60. For each day that he does not work, \$30 is subtracted from his earnings. At the end of the 20-day period, he received \$660. How many days did he not work? Let's think step by step and output the final answer within $\square$. | 6 | 1.00 |
| Top-2 | 10 | The points $(9, -5)$ and $(-3, -1)$ are the endpoints of a diameter of a circle. What is the sum of the coordinates of the center of the circle? Let's think step by step and output the final answer within $\square$. | 0 | 1.00 |
| Top-3 | 15 | Let $f(x) = \begin{cases} x/2 & \text{if } x \text{ is even,} \\ 3x + 1 & \text{if } x \text{ is odd} \end{cases}$. What is $f(f(f(f(1))))$? Let's think step by step and output the final answer within $\square$. | 4 | 1.00 |
| Top-4 | 17 | Let $f(x) = \begin{cases} 2x^2 - 3 & \text{if } x \leq 2, \\ ax + 4 & \text{if } x > 2 \end{cases}$. Find $a$ if the graph of $y = f(x)$ is continuous. Let's think step by step and output the final answer within $\square$. | $\frac{1}{2}$ | 1.00 |
| Top-5 | 32 | Simplify $(2x - 5)(x + 7) - (x + 5)(2x - 1)$. Let's think step by step and output the final answer within $\square$. | -30 | 1.00 |
| **Bottom-5 Examples** | | | | |
| Bottom-1 | 6 | What are all values of $p$ such that for every $q > 0$, we have $\frac{3(pq^2 + p^2q + 3q^2 + 3pq)}{p+q} > 2p^2q$? Express your answer in interval notation in decimal form. Let's think step by step and output the final answer within $\square$. | $[0, 3)$ | 0.00 |
| Bottom-2 | 39 | The square of an integer is 182 greater than the integer itself. What is the sum of all integers for which this is true? Let's think step by step and output the final answer within $\square$. | 1 | 0.00 |
| Bottom-3 | 48 | Find the product of all constants $t$ such that the quadratic $x^2 + tx - 10$ can be factored in the form $(x + a)(x + b)$, where $a$ and $b$ are integers. Let's think step by step and output the final answer within $\square$. | 729 | 0.00 |
| Bottom-4 | 49 | Factor $58x^5 - 203x^{11}$. Let's think step by step and output the final answer within $\square$. | $-29x^5(7x^6 - 2)$ | 0.00 |
| Bottom-5 | 7487 | Compute $\cos 72°$. Let's think step by step and output the final answer within $\square$. | $\frac{-1+\sqrt{5}}{4}$ | 0.00 |

Table 11: Details of Top-5 and Bottom-5 examples ranked by the combined scoring (the proposed Star-DS method).

| Rank | Index | Prompt | Ground Truth | Score |
|------|-------|--------|--------------|-------|
| **Top-5 Examples** | | | | |
| Top-1 | 596 | Let $f(x) = \begin{cases} -x+3 & \text{if } x \leq 0, \\ 2x-5 & \text{if } x > 0 \end{cases}$. How many solutions does the equation $f(f(x)) = 4$ have? Let's think step by step and output the final answer within $\square$. | 3 | 2.00 |
| Top-2 | 3963 | Let $a, b, c$ be nonzero real numbers, and let $x = b/c + c/b, y = a/c + c/a, z = a/b + b/a$. Simplify $x^2 + y^2 + z^2 - xyz$. Let's think step by step and output the final answer within $\square$. | 4 | 2.00 |
| Top-3 | 7197 | Let $l, m, n$ be real numbers, and let $A, B, C$ be points such that the midpoint of $\overline{BC}$ is $(l, 0, 0)$, the midpoint of $\overline{AC}$ is $(0, m, 0)$, and the midpoint of $\overline{AB}$ is $(0, 0, n)$. Find $\frac{AB^2 + AC^2 + BC^2}{l^2 + m^2 + n^2}$. Let's think step by step and output the final answer within $\square$. | 8 | 2.00 |
| Top-4 | 500 | A 100-gon $P_1$ is drawn in the Cartesian plane. The sum of the $x$-coordinates of the 100 vertices equals 2009. The midpoints of the sides of $P_1$ form a second 100-gon, $P_2$. Finally, the midpoints of the sides of $P_2$ form a third 100-gon, $P_3$. Find the sum of the $x$-coordinates of the vertices of $P_3$. Let's think step by step and output the final answer within $\square$. | 2009 | 1.98 |
| Top-5 | 258 | The entire graph of the function $f(x)$ is shown below ($f$ is only defined when $x$ is between -4 and 4 inclusive). How many values of $x$ satisfy $f(f(x)) = 2$? [asy]...[/asy] Let's think step by step and output the final answer within $\square$. | 3 | 1.94 |
| **Bottom-5 Examples** | | | | |
| Bottom-1 | 4662 | The complex number $z$ traces a circle centered at the origin with radius 2. Then $z + \frac{1}{z}$ traces a: (A) circle (B) parabola (C) ellipse (D) hyperbola. Enter the letter of the correct option. Let's think step by step and output the final answer within $\square$. | C | 0.00 |
| Bottom-2 | 4988 | A school has between 150 and 200 students enrolled. Every afternoon, all the students come together to participate in gym class. The students are separated into six distinct sections of students. If one student is absent from school, the sections can all have the same number of students. What is the sum of all possible numbers of students enrolled at the school? Let's think step by step and output the final answer within $\square$. | 1575 | 0.00 |
| Bottom-3 | 7103 | Find the curve defined by the equation $r = 4 \tan \theta \sec \theta$. (A) Line (B) Circle (C) Parabola (D) Ellipse (E) Hyperbola. Enter the letter of the correct option. Let's think step by step and output the final answer within $\square$. | C | 0.00 |
| Bottom-4 | 7191 | For a positive constant $c$, in spherical coordinates $(\rho, \theta, \phi)$, find the shape described by the equation $\rho = c$. (A) Line (B) Circle (C) Plane (D) Sphere (E) Cylinder (F) Cone. Enter the letter of the correct option. Let's think step by step and output the final answer within $\square$. | D | 0.00 |
| Bottom-5 | 7371 | Find the curve defined by the equation $r = \frac{1}{1 - \cos \theta}$. (A) Line (B) Circle (C) Parabola (D) Ellipse (E) Hyperbola. Enter the letter of the correct option. Let's think step by step and output the final answer within $\square$. | C | 0.00 |