# OpenReview forum: "Star-DS: Step-level Uncertainty-Aware Reasoning Data Selection in Reinforcement Learning for LLM Multi-step Reasoning"
_ICLR.cc/2026/Conference — ICLR 2026 Conference Withdrawn Submission_

### Official Review · Reviewer_4F6B · 2025-10-28

**Soundness:** 2
**Presentation:** 2
**Contribution:** 2
**Rating:** 2
**Confidence:** 4

**Summary:**

The paper proposes STAR-DS, a data-selection method targeting multi-step reasoning RL for large language models. It argues that existing methods focus only on outcome-level signals (final answer correctness) and ignore step-level uncertainties in the reasoning chain. STAR-DS computes a composite score combining (a) step-wise self-evaluation uncertainty and (b) outcome-level reward variability. It then uses the top-K highest-uncertainty samples to fine-tune via RL (e.g., GRPO) on multi-step reasoning benchmarks, showing that using far fewer training examples selected by STAR-DS can match or exceed full-dataset training while reducing compute/data cost.

**Strengths:**

1. Good motivation to tackle large RL training cost for multi-step reasoning tasks.

2. Experiments show efficiency gains: with only ~1,000 selected examples they achieve near/full dataset performance.

**Weaknesses:**

1. Outcome-level uncertainty or question difficulty has already been widely used for prompt selection and filtering in prior work ([1] [2]). This work mainly adds step-level uncertainty on top of previous idea, but at the cost of significantly increased computational overhead and system complexity. This raises questions about whether the additional signal justifies the extra cost.

2. Limited gains from step-level uncertainty: As shown in Table 2, the stepwise uncertainty signal alone performs worse than reward variability, and even when combined, the average gain is only around 0.5 points. Considering the added complexity, the marginal benefit seems limited.

3. Incremental improvement over simpler baselines:
When compared to much simpler baselines such as LIM or IFD, the performance improvement is small. For example, as shown in Table 6 (100-sample setting), XRPO performs close to LIM/IFD, suggesting that the added step-level uncertainty may not yield a clear advantage in practice.

4. Limited model scale:
All experiments are conducted only on 1.5B models. The generality and scalability of the approach on larger models remain unclear.

[1] Process Reinforcement through Implicit Rewards
[2] DAPO: An Open-Source LLM Reinforcement Learning System at Scale

**Questions:**

Please see above weaknesses.

---

### Official Review · Reviewer_6WcF · 2025-11-02

**Soundness:** 3
**Presentation:** 3
**Contribution:** 2
**Rating:** 4
**Confidence:** 4

**Summary:**

This paper proposes a novel Step-level Uncertainty-Aware Reasoning Data Selection approach (Star-DS) that incorporates both step-level and outcome-level signals for identifying high-value reasoning data in reinforcement learning for LLM reasoning. Specifically, the authors introduce step-wise self-evaluation uncertainty of each reasoning step, as well as reward variance of the final answer, to quantify the value of each sample. Experiments show Star-DS consistently identifies high-value data, significantly reduces both data requirements and computational costs.

**Strengths:**

1. This paper is well structured and easy to follow.

2. This paper integrates step-level self-evaluation uncertainty for data selection, overcoming the shortcomings of previous approaches that rely solely on outcomes.

3. The results outperform previous selection methods like LIM, which demonstrate the effectiveness of proposed process signals.

**Weaknesses:**

### Method
- The authors adopt an offline data selection strategy before RL training, which may not adapt to the dynamics of learning. It's uncertain whether the initially selected data will retain the high value as the model is updated.

- The authors adopt self-evaluation to calculate sample-level uncertainty. But I notice that Qwen2.5-Math-1.5B is used in practice. It's not clear whether a robust judgement of the correctness of responses can be made. I suggest providing comparisons with larger, advanced models to validate whether there are significant differences in this metric.

### Experiments
- The model is limited to the Qwen2.5 series. While the Qwen2.5 series is well-pretrained to provide a solid foundation in post-training, it also raises concerns on data contamination in widely used benchmarks [1]. Consequently, breakthroughs are predominantly observed for the mathematically strong Qwen2.5 series on benchmarks such as MATH-500, AMC, and AIME, and seldom transfer to models like Llama. I believe a more in-depth investigation on other model families (e.g., Llama) is needed to validate the effectiveness of Star-DS.

- Concerns on computational efficiency. The authors briefly discuss the computational costs in Section 4.2. It seems that it's more costly compared to the full dataset when the training epoch is less than 10. I suggest providing more discussion on the computational efficiency regarding other baselines, such as GRESO, LIM, etc.

---
[1] Reasoning or Memorization? Unreliable Results of Reinforcement Learning Due to Data Contamination. arXiv preprint arXiv:2507.10532

**Questions:**

1. 1-shot RLVR [1] uses only *1 training example* and is effective in incentivizing the reasoning capabilities of LLMs. It sustained test performance improvement even after the training accuracy has saturated. I noticed that Star-DS selects *1,000 example* for training. How does its performance compare to 1-shot-RLVR?

2. Why choose the pattern of "LLM-as-a-Judge" to calculate the uncertainty? Why not use the confidence inside the models as metrics?

3. Methods like GRESO and LIM consider learning dynamics, and GRESO dynamically selects data during training. The proposed method uses offline selection. Would online data selection that combines training dynamics result in better performance?

4. I notice that Star-DS outperforms LIM by only using outcome reward variability in Tables 1,2. Since LIM not only uses outcome reward but also considers training dynamics, the even worse results are very confusing to me. Can the authors provide an explanation? Why does incorporating training dynamics reduce performance?

---

[1] Reinforcement Learning for Reasoning in Large Language Models with One Training Example. Wang et al., 2025

---

### Official Review · Reviewer_e7xC · 2025-11-02

**Soundness:** 1
**Presentation:** 3
**Contribution:** 1
**Rating:** 0
**Confidence:** 4

**Summary:**

This paper proposes a data selection method for LLM reasoning RL, by measuring uncertainty from two complementary perspectives: step-wise uncertainty, which captures instability within intermediate reasoning steps, and reward variability, which quantifies divergence across final outcomes.

**Strengths:**

1. The paper is clearly written and easy to follow.
2. The studied problem, i.e., using as few data as possible for LLM reasoning training, is important.

**Weaknesses:**

1. The uncertainty quantification procedure requires a self-evaluation step. It might be inaccurate if the model is not trained to do self-rewarding. There are also no ablation studies on the accuracy of self-evaluation and its impact on the RL training.
2. Besides, the benefit of the proposed method can be attributed solely to the Reward Variability term, which essentially removes the data that is all correct or all wrong. GRPO will give no gradient for such data, which may lead to slower convergence and lower performance compared to the proposed method. However, the proposed method does not save much computation since rollouts are still needed for each prompt before dropping these invalid samples.
3. The experiments also cannot fully support the conclusion. Only a small size (1.5B) Qwen model is evaluated, which is not enough to show the effectiveness of the method. And the results are pass@1. For AIME and AMC, where the number of problems is few, pass@1 might be a noisy metric, compared to pass@N or avg@N.
4. The setting is also somewhat weird. The experiments adopt GRPO for 200 epochs on 1,000-example subsets of MATH. It is not a regular setup since too few samples and too many epochs are performed, which can lead to overfitting and the learning of only format during GRPO.

**Questions:**

See weakness.

---

### Official Review · Reviewer_MqK2 · 2025-11-10

**Soundness:** 2
**Presentation:** 2
**Contribution:** 1
**Rating:** 2
**Confidence:** 4

**Summary:**

The paper presents Star-DS, a data selection method when doing RL on reasoning LMs. For each training instance, the method combines two signals to select rollouts: a step-level uncertainty score derived from the model’s own evaluation (prompting itself) is combined together with outcome rewards. After normalizing these components, their sum is used to rank and select the top-K samples for training.
The authors evaluate this approach with small Qwen2.5-Math-1.5B and a distilled qwen on math benchmarks.

**Strengths:**

1. The paper is easy to read.
2. Using step-level signals is a reasonable direction given that multi-step reasoning contains rich internal structure. The combination of step-wise uncertainty and reward variability is simple but intuitive.

**Weaknesses:**

1. The approach leans heavily on self-evaluation prompts, which are known to be unstable and sometimes misleading. A model judging its own steps can blur the line between actual reasoning quality and self-consistency bias, so the “uncertainty” signal may not reflect genuine difficulty.

2. Uncertainty calibration is questionable.

3. The final 'sum' scoring is quite ad-hoc: normalize two unrelated numbers and add them. There is no clear justification for why this linear combination should correspond to training value.

4. While the experiments are broad, most gains are small (often within 1–2%). These deltas are within typical random noise for math-reasoning evaluations.

5. empirically: settings and models are very limited. Using Qwen3 for training on a few math datasets is questionable. two groups of experiments, each using only one different backbone.

6. The cost of computing step-level uncertainty is also significant. you need to account for generating multiple rollouts and running step-wise evaluations.

**Questions:**

Does Star-DS actually identify samples that help the model learn, or does it simply prefer prompts that trigger inconsistent rollouts?

---

### Note · Authors · 2025-11-26

**Comment:**

We sincerely thank all reviewers for their thoughtful and constructive comments. We truly appreciate the time and effort invested in reviewing our work. We will carefully consider all feedback in preparing a revised version.

**Withdrawal Confirmation:**

I have read and agree with the venue's withdrawal policy on behalf of myself and my co-authors.